# Geo-Questionnaire for Environmental Planning: The Case of Ecosystem Services Delivered by Trees in Poland

**Patrycja Przewoźna [1,2], Adam Inglot [3,*], Marcin Mielewczyk [2], Krzysztof Mączka [2] and Piotr Matczak [2] and Piotr Wężyk [4]**

1   Department of Geoinformation, Faculty of Geographical and Geological Sciences, Adam Mickiewicz University, ul. Wieniawskiego 1, 61-712 Poznan, Poland; pwysocka@amu.edu.pl
2   Department for the Study of Social Dynamics, Faculty of Sociology, Adam Mickiewicz University, ul. Wieniawskiego 1, 61-712 Poznan, Poland; marmie@st.amu.edu.pl (M.M.); krzysztof.maczka@amu.edu.pl (K.M.); matczak@amu.edu.pl (P.M.)
3   Department of Geodesy, Faculty of Civil and Environmental Engineering, Gdansk University of Technology, ul. Narutowicza 11/12, 80-233 Gdansk, Poland
4   Department of Forest Resources Management, Faculty of Forestry, University of Agriculture in Krakow, Al. 29 Listopada 46, 31-425 Kraków, Poland; p.wezyk@ur.krakow.pl
*   Correspondence: adam.inglot@pg.edu.pl

**Abstract:** Studies on society and the environment interface are often based on simple questionnaires that do not allow for an in-depth analysis. Research conducted with geo-questionnaires is an increasingly common method. However, even if data collected via a geo-questionnaire are available, the shared databases provide limited information due to personal data protection. In the article, we present open databases that overcome those limitations. They are the result of the iTre-es project concerning public opinion on the benefits provided by trees and shrubs in four different research areas. The databases provide information on the location of trees that are valuable to the residents, the distances from the respondents' residence place, their attitude toward tree removal, socio-demographic variables, attachment to the place of life, and environmental attitudes. The presentation of all these aspects was possible thanks to the appropriate aggregation of the results. A method to anonymize the respondents is presented. We discuss the collected data and their possible areas of application.

**Dataset:** The effect of interview location on the perception of Ecosystem Services provided by trees. A Polish case study. DOI: 10.34808/apm8-re13; Attitudes to tree removal on private property in rural and urban Polish municipalities. DOI: 10.34808/ds29-zt75; Attitudes to tree removal on private properties in two Polish cities. DOI: 10.34808/6d3c-qy88.

**Keywords:** PPGIS; greenery management; environmental sociology

## 1. Summary

In geoinformation, one of the most promising approaches developed since the 1990s is Public Participation Geographic Information Systems (PPGISs), which allow inhabitants to be included in the decision-making process using maps. Different approaches enable the practical implementation of this idea, including web-based methods such as geo-questionnaires [1]. This belongs to the Computer-Assisted Web Interviewing (CAWI) group of methods based on the softGIS methodology, which concentrates on collecting spatial information in a user-friendly way with interactive maps [2]. PPGISs provide a methodology for many applications related to the effective management of urban spaces [3], including environmental planning [4].

Geo-questionnaires are one of the methods recommended for mapping ecosystem services (ESs)—direct benefits that the natural environment delivers to people and that positively affect human well-being [5]. Geo-questionnaires are indispensable, mainly when intangible benefits, such as recreation or aesthetic value, are analyzed [6]. ESs provided by trees are one of most critical, especially in urban areas [7]. The most extensive research in this field was carried out in Poland as part of the "Count on greenery" project, but it focused only on ES related to recreation. Moreover, the report and the provided spatial data focus on green spaces without reference to the respondents participating in the survey. https://uslugiekosystemow.pl/baza-wiedzy/badania-analizy-i-raporty/licz-na-zielen-mapy-z-wynikami-badan/ (accessed on 1 October 2019).

However, an in-depth understanding of both the spatial diversity of mapped ESs and the motivations and demographic profiles of the respondents using them is crucial for effective urban green management [4,8]. To address this challenge, we designed an extensive exploratory project utilizing both types of data, objectified (derived from LiDAR or geolocation) and originating from sociological (quantitative and qualitative) methods. The geo-questionnaire in this article presents the quantitative data that were collected within the project "iTre-es—the impact of institutional framework change on ecosystem services provided by trees/shrubs to local communities" (funded by the National Science Centre in Poland within a research grant under the number 2017/25/B/HS6/00954). The questionnaire itself was subsequently used thanks to the financial support of the Faculty of Geographical and Geological Sciences at Adam Mickiewicz University in Poznań.

The goal of the study was to indicate ecosystem services that the inhabitants of the research areas appreciated and the factors influencing their perception. We collected data on the location of trees of high value to the residents, including the benefits perceived by them (regarding 17 different ESs), and much information about the respondents themselves, enabling in-depth investigation. The quantitative data then allowed us to conduct analyses to compare views of urban green management with the variable effect of place attachment [9], as well as the usefulness of location data from PPGISs (e.g., geo-questionnaire) for comparisons with LiDAR data in subsequent articles [10,11]. This article briefly depicts how we conducted the survey and aggregated the data and then discusses the responses and potential methods of using them for subsequent data compilations. We made the collected data available in a public repository with open access: GDAPOZ19 [12], RACNYS19 [13], GDAPOZ20 [14].

## 2. Data Collection

The geo-questionnaire contained seven sections with topics for which respondents could express their opinions about trees in their vicinity and provide information about themselves and their place of residence:

Section 1—Data sheet—information about the respondent:

Data included information on gender, age, and education. All of these questions were single-choice questions. Moreover, one multiple-choice question was also asked about the respondent's occupation.

Section 2—Place of residence:

This section contains a question about the place of residence. Using a digital map, the respondent inserted a point specifying the place of residence and answered questions about the type of development and whether there was a tree in the vicinity of the residence.

Section 3—Trees in the area:

In this section, respondents were asked to indicate a tree or trees beneficial to them. This was possible by inserting a point on the map where the tree or trees were located. For each localization, the respondent assigned one or more ESs from the drop-down menu list of 17 predefined ESs (Table 1) [15,16]. Moreover, respondents were asked the type of land ownership where the tree or trees were located. After approval of one tree, respondents could select another tree or go to the next section.

Section 4—New trees proposed by respondents:

In this section, respondents were asked to express their opinions on the need to plant a new tree by inserting a point on the map where the new tree should be located. Furthermore, respondents were also asked to assign the reason from the list of seven available answers. It was the third and final question using geolocation in the questionnaire.

Section 5—Respondents attitudes toward removing trees:

This section included two questions: (1) Who should decide on tree removal—the municipality administration or the owner of the land (a single-choice choice questions with three predefined answers and respondents own answer? (2) For what reasons does the respondent agree with cutting trees (a multiple-choice question with a predefined list of 11 answers)?

Section 6—Place attachment

This section was related to the respondent's attachment to the place of residence. The 9 questions scale of place attachment was applied and respondents answered using five level Likert item from "I strongly disagree" to "I strongly agree".

Section 7—New Ecological Paradigm:

This section included the 15-question scale of New Ecological Paradigm (NEP) concerning attitudes towards the environment [17]. These questions constitute a standard for determining the respondent's environmental awareness.

**Table 1.** List of 17 ES benefits based on the TEEB [18] classification and Kronenberg's research [16].

| Short | Ecosystem Services |
|---|---|
| ES1 | Noise reduction |
| ES2 | Educational usefulness (e.g., nature lessons in outdoors) |
| ES3 | Air and soil humidification |
| ES4 | Supplying wood, branches, and leaves |
| ES5 | Impact on the aesthetics of space |
| ES6 | Positive impact on health and well-being |
| ES7 | Delivery of fruit and nuts |
| ES8 | A sense of intimacy, separating from neighbors |
| ES9 | Sun protection (shadow) |
| ES10 | Strengthening interpersonal bonds, psychological relationship between people and trees, sense of attachment to the place (personal experience) |
| ES11 | Oxygen source |
| ES12 | Air purification |
| ES13 | Place of life of animals and their source of food |
| ES14 | Place of recreation |
| ES15 | Wind protection |
| ES16 | The tree as a witness to history: trees aged several hundred years, bearing traces of events, important for regional heritage |
| ES17 | Protection against snowdrifts |

The data produced as part of the research covered three collections: GDAPOZ19 [12], RACNYS19 [13], GDAPOZ20 [14]. Table 2 presents the names of the fields in the shared datasets. These fields were classified according to the form page or the metadata class. Each field was accompanied by a short description and scope of data for the selected fields. Detailed information was placed in an additional file with the questions and answers that the respondent had at his/her disposal.

All datasets are made available in Microsoft Excel format with the xlsx extension; this format is convenient for users of the Pandas module and for people using a spreadsheet. The datasets formed a table with the following sizes: GDAPOZ19 ($118 \times 251$), RACNYS19 ($371 \times 535$), GDAPOZ20 ($2420 \times 1570$). Due to the size of the GDAPOZ20 table, we recommend analyzing the data in the Pandas module. The difference in the number of fields resulted from the different maximum number of trees selected by the respondents (value "n" in Table 2). A lack of data (no answer indicated by the respondent) in the xlsx

file was entered with an empty cell, while in the Pandas module, it is represented by the value "e.g., nan."

**Table 2.** Description of the data contained in the datasets, distinguishing between fields appearing in the datasets.

| Class of Field | Field Name in Database | Description |
|---|---|---|
| metadata | LocalID | Local identifier, unique within a single geo-questionnaire: GDAPOZ19 (1-118), RACNYS19 (1-371), GDAPOZ20 (1-2420) |
| | SID | A global identifier, unique across all three datasets |
| | City | Acronyms of the town: G—Gdańsk, P—Poznań, N—Nysa, R—Racibórz (no field in GDAPOZ19) |
| | CityIOid | Acronyms of the town: G—Gdańsk, P—Poznań, together with the location of the interview: O—outside and I—inside (only in GDAPOZ19) |
| | Date | Date of the interview: GDAPOZ19 (2019.04.09:2019.05.13), RACNYS19 (2019.05.10:2019.07.12), GDAPOZ20 (2020.06.22:2020.08.04), date format: yyyy.mm.dd |
| | Time | Time of starting the geo-questionnaire |
| | TimeFT | This field stores the time of completing the survey; the average time for each dataset is: GDAPOZ19 (00:11:11), RACNYS19 (00:23:15), GDAPOZ20 (00:17:30) |
| data sheet (Section 1) | Sex | Basic information about the respondent |
| | Age | |
| | Edu | |
| | Work1—Work8 | |
| attitude toward felling trees (Section 5) | Q1 | Question—"Who, according to your opinion, should decide about trees removal on private possessions?" |
| | Q2 | Open question—"In which cases should the owner be able to decide about trees removal on his property?" |
| | Q3 | Multiple choice question "I would decide to remove the tree on my property if ..." |
| place attachment (Section 6) | PA1–PA9 | Nine questions related to the respondent's attachment to the indicated place of residence |
| | MRPA | The incomplete number of the nine questions about place attachment |
| | PAV | Mean value of the answers PA1–PA9 taking into account the context of the question (PA2 and PA6—negation) |
| NEP (Section 7) | EPQ1–EPQ15 | Percentage of geo-questionnaire with all answers: GDAPOZ19 (8%), RACNYS19 (19%), GDAPOZ20 (12%); percentage of geo-questionnaire with at least half of the answers: GDAPOZ19 (94%), RACNYS19 (58%), GDAPOZ20 (46%) |
| | Q3 | Multiple choice question "I would decide to remove the tree on my property if ..." |
| place of residence (Section 2) | DT_address | Date and time of opening the question with an indication of the place of residence |
| | POINT_X_address | The incomplete number of the nine questions about place attachment |
| | POINT_Y_address | |
| | Q4 | Question about the type of building in which the respondent lives |
| | Q5 | Are there any trees in the immediate vicinity? |

**Table 2.** *Cont.*

| Class of Field | Field Name in Database | Description |
| --- | --- | --- |
| beneficial trees (Section 3) | NumberBT | Number of indicated beneficial trees: GDAPOZ19 (197), RACNYS19 (401), GDAPOZ20 (2296) |
| | nESDT | Date and time of opening the question about indicated beneficial trees |
| | POINT_X_nES | Coordinates of the hexagon centroids in which the indicated beneficial trees are located |
| | POINT_Y_nES | |
| | Dis_nES | The selected ecosystem service for the indicated tree or trees; n is the maximum number of locations indicated by the respondents: GDAPOZ19 (n = 6), RACNYS19 (n = 9), GDAPOZ20 (n = 13) |
| | nESLoc | Where and on what terrain the indicated tree or trees are located |
| new trees (Section 4) | NumberNT | Number of new trees indicated: GDAPOZ19 (139), RACNYS19 (305), GDAPOZ20 (2038) |
| | nNESDT | Date and time of opening the question about new trees |
| | POINT_X_nNES | Coordinates of the hexagon centroids in which the indicated new trees are located |
| | POINT_Y_nNES | |
| | Dis_1NES | Distance of the indicated tree from the indicated place of residence |
| | nNES1 – nNES7 | The selected ecosystem service for the indicated single new tree; n is the maximum number of new trees indicated by the respondents: GDAPOZ19 (n = 5), RACNYS19 (n = 22), GDAPOZ20 (n = 111) |

## 3. Methods

### 3.1. Research Area

Our research was conducted in four areas (Figure 1A) in 2019 and 2020 (Table 2). This included:

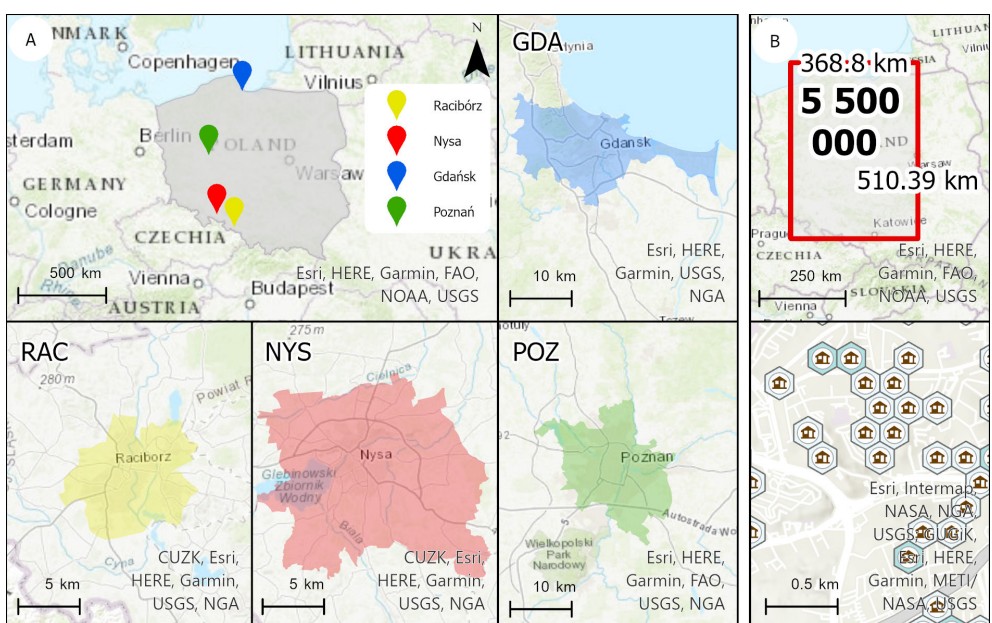

**Figure 1.** Location of (**A**) the case study areas and (**B**) the extent and distribution of hexagons for the GDAPOZ20 data [14].

- A pilot survey among groups of students from Poznań and Gdańsk testing the geo-questionnaire and the method (especially the impact of the place where the respondents completed the questionnaire; therefore, the testing took place indoors and outdoors);

- Comparative research of the urban municipality (Racibórz) and the rural one (Nysa, which is an urban–rural municipality in which we focused on three selected villages: Sękowice, Regulice, Konradowa);
- Research concentrated on the inhabitants of large cities in Poznań and Gdańsk.

### 3.2. Respondents Recruitment

In order to recruit respondents, several methods previously used in PPGIS applications [1] were applied. They included random samplings with unaddressed mail invitations sent to households supported with field research and voluntary sampling (through local press and social media). However, social media invitations were the most effective in terms of response rate (Table 3). The questionnaire was answered online following the survey described above.

In the pilot survey, students were selected from studies unrelated to biological or forestry sciences for the pilot study, avoiding the influence of respondents' education on the obtained results. The research was conducted in the presence of a research team representative of two groups. The first one was carried out in a greenery area near the Gdańsk University of Technology and the Poznań University of Technology. The second one was in the computer lab.

**Table 3.** The basic characteristics of the case study areas and information about the data collection.

| Municipality | Type | Area km$^2$ | Population | Data Collection Period | Ways of Obtaining Respondents |
|---|---|---|---|---|---|
| Racibórz (RAC) | urban | 75 | 55,000 | late spring and summer 2019 | unaddressed mail invitations sent to households; invitations through local press and on social media (Facebook); field research |
| Nysa (NYS) | urban-rural | 218 | 57,000 | late spring and summer 2019 | unaddressed mail invitations sent to households; invitations through local press and on social media (Facebook); field research |
| Poznań (POZ) | urban | 262 | 533,000 | spring 2019 | volunteer student recruitment |
| | | | | summer 2020 | invitations through the local press and on social media (Facebook) |
| Gdańsk (GDA) | urban | 265 | 582,000 | spring 2019 | volunteer student recruitment |
| | | | | summer 2020 | invitations through the local press and on social media (Facebook) |

The respondents recruited for the study in Nysa and Racibórz were obtained by sending non-addressed invitations by traditional mail, information in the local press, and social media. However, it was also necessary to attract inhabitants of both areas to participate by applying tablet surveys in the field. It was necessary to increase the number of respondents and improve their representativeness, especially in the case of the rural municipality of Nysa.

In the case of the field research, the respondents filled in the questionnaire independently, and the research team representative intervened only at the respondent's request.

In the case of the research carried out in Gdańsk and Poznań, recruitment was limited to acquiring respondents with the help of the local press and social media (paid advertising on Facebook). This was due to the precautionary measures in connection with the COVID-19 pandemic and the effectiveness of this method of acquiring respondents, observed during the research carried out in the city of Racibórz.

### 3.3. Aggregation of the Results

The data collected via the geo-questionnaire included the location of respondents' residences. In order to be published as open research data, the location data of address

points and designated trees had to be anonymized. Maintaining anonymity involved representing the original data by the polygon grid (triangles, squares, or hexagons) or the centroid of these polygons. Matching the regular size is challenging because a compromise must be made between maintaining data detail and anonymity. The minimum base field size at which respondent anonymity would be preserved is required to determine the compromise. As a basis for ensuring anonymity, at least two apartments were used in each tessellation, which gave us the hexagon area of 3.41 ha.

The second problem in building anonymized data is the data layer extent. Making a shape grid of regular hexagons for the given range would result in the layer containing more than five million hexagon tessellations (Figure 1B). Conducting calculations on such a layer produces disproportionate results to the time required to perform analytical work in the program. We prepared a script that builds a layer only from those tessellations that contain selected locations to solve this problem. The choice of hexagonal tessellation was due to the properties of hexagons. They keep the distance between all neighboring cells constant, which is essential when considering the neighborhood of objects [19].

## 4. User Notes

The data we collected are stored in three open databases: GDAPOZ19 [12]—research with students, RACNYS19 [13]—research with residents of Nysa and Racibórz in 2019, and GDAPOZ20 [14]—research with the residents of Gdańsk and Poznań in 2020. The first one is already available online; the other two will be available by the end of the year 2021. The number of complete questionnaires collected in Gdańsk and Poznań (address point located in GDA = 997, POZ = 810) is comparable to the previous experience of this type of research in Poland [1,20]. Nevertheless, the number of responses obtained in Racibórz (165 responses) was much under our expectations despite greater involvement in collecting data (considering various methods of reaching respondents). More inhabitants took part in surveys realized in municipalities of a similar size as Rokietnica or Swarzędz, localized in the vicinity of Poznań [20]. Racibórz, however, is not a suburban municipality, which may have a significant impact on the involvement of residents in public participation. In the case of Nysa, 104 questionnaires were obtained, which should be considered a satisfactory result. According to the information obtained from the municipality office, 10% of all people living in the villages were surveyed.

Nevertheless, the overall response of the inhabitants of both municipalities confirmed that geo-questionnaires are more efficient in involving people from big cities than small urban or rural areas [4]. In this context, open databases are even more crucial because collecting reliable information from such areas is still a challenging task. Open data make it possible to collect comparable information from different areas with similar characteristics, allowing for in-depth research using geo-questionnaires regarding ESs provided by trees in these types of municipalities.

It needs to be highlighted that the collected datasets are not representative of the populations. This is a well-known problem with the discussed data collection method [1]. In the case of both datasets RACNYS19 [13] and GDAPOZ20 [14], we observed the over-representation of young people (Figure 2), similarly to Bąkowska et al. [20]. In Nysa and Poznań, 42% of respondents represented a higher level of education; in Poznań and Gdańsk, this was 57%. At the same time, according to the national census, the share of this group should be 12% and 30%, respectively. Similar problems occurred previously in research applying geo-questionnaires [4,20,21]. The overrepresentation of people with a higher income also occurred in the past [4,21]. Thus, this may also be the case of the discussed datasets, although this indicator was not controlled in our study. Interestingly, we also observed a more significant share of women than men, which may indicate their high involvement in this topic. So far, a greater share of men than women has been reported [21] or an equal share of both males and females [20].

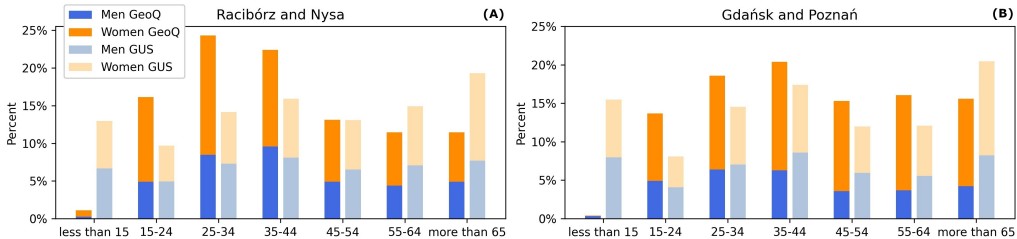

**Figure 2.** Summary of the age structure of respondents (GeoQ—blue bars on the chart) from (**A**) Racibórz and Nysa [13] and (**B**) Gdańsk and Poznań [14] with Statistics Poland data (GUS—orange bars on the chart) available on Local Data Bank (https://bdl.stat.gov.pl/BDL, accessed on 1 November 2019).

Nevertheless, this does not preclude the practical applicability of such data. It only underlines the need to supplement online data collection by methods based on direct contact with respondents. Earlier research indicated that both solutions, online and face-to-face, should be used simultaneously [4,20], especially since the web-based GIS methods, such as geo-questionnaires, allow reaching a larger group of potentially interested recipients [3,20] and engaging new groups of participants [3].

**Author Contributions:** Conceptualization, P.P., A.I., P.M.; methodology, P.P., A.I., M.M., K.M., P.M., P.W.; validation, P.P., A.I., P.W.; formal analysis, P.P., A.I.; resources, A.I., M.M.; data curation, A.I.; writing—original draft preparation, P.P., A.I., M.M., K.M., P.M.; writing—review and editing, P.P., A.I., P.W.; visualization, P.P., A.I.; supervision, P.M., P.W.; funding acquisition, P.P., K.M., P.M., P.W. All authors have read and agreed to the published version of the manuscript.

**Funding:** This research was funded by the National Science Centre, Poland, under grant number 2017/25/B/HS6/00954.

**Data Availability Statement:** The datasets presented in this study have been released with the doi (10.34808/apm8-re13; 10.34808/ds29-zt75; 10.34808/6d3-qy88.) in MOST Wiedzy under CC BY license.

**Conflicts of Interest:** The authors declare no conflict of interest.

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
