# Peer review of "Geo-Questionnaire for Environmental Planning: The Case of Ecosystem Services Delivered by Trees in Poland"

_data, 2021_

Round 1

Reviewer 1 Report

Dear Authors,

I have read your manuscript and I have 3 issues you should address:

  1. in the abstract it would be nice to make it clearer what the purpose of the research was and what the purpose of the present paper is;
  2. In the manuscript (Lines 146-147) you are saying that "A minimum of two dwellings in each tessellation was used as the basis for securing anonymity." In the database you are referring to (https://mostwiedzy.pl/en/open-research-data/attitudes-to-tree-removal-on-private-properties-in-two-polish-cities,607060449649287-0) you are saying that "Data are presented in aggregate form in hexagonal units with an area of approx 34,6 m2." I think it is not enough for a two dwelling distance . . . Please clarify!
  3. The English must be checked thoroughly. I marked some of the issues but you really need to check the whole text.

Please check the pdf with my comments and decide what you change.

Best regards, Reviewer X

Author Response

Answer 1: Thank you for noting. We have included the purpose of the research in the summary section and underline it better in the abstract.

Answer 2: Thank you for your attention. In fact, the error occurred, the hexagon area is 3.46 [ha] and not 34.6 [m2]. We included the correct area in the text and added a note in the README file in the data.

Answer 3: Thank you for all remarks and careful reading of the text! We included your suggestions. We have also improved the overall text in terms of style and language.

Reviewer 2 Report

I think that the authors should explain the relevance of the database created. It seems that some of the ecosystem services mentioned in tab 1 , can not be evaluated from the database created.....for example : E1, E15, E 17 etc. can not be evaluated from a simple tree. It is good for  E 16, E 10, E2 etc. So, I think that a paragraph can explain the purpose of the database and the use of it. It is sufficient for the publication of the paper. Thank You!

Author Response

Answer: Thank you for this remark! We have included the purpose of the research in the summary section. Nevertheless, this confusion is also related to a lack of precision during the question translation. Respondents were asked to mark the localization of tree or trees that are beneficial to them. We have refined this point in the text as well. Since marking the whole area is a much more difficult task for non-skilled map users, we limited the question to localization of beneficiary trees or trees with a simple point in both cases.  An unambiguous interpretation of this selection is possible thanks to the attached information about the type of land on which the tree / trees is located (nESLoc item in the metadata description)

Reviewer 3 Report

The paper presents a stimulating research which has been conducted in Poland by means of geo-questionnaires with a method to anonymize the respondents.

The following revisions and suggestions are recommended:

  • Figure 1 can be enlarged to better read the images.

  • Lines 161-171: The responses obtained in Racibors (165 responses) and Nysa (104 responses) are indicated but the questionnaires obtained in the two other locations (Gdansk and Poznan) are not indicated. Insert this information.

Author Response

Answer 1: The figure 1 has been corrected

Answer 2: The note has been supplemented in the text